# Tumor-Targeted Drug Conjugates as an Emerging Novel Therapeutic Approach in Small Cell Lung Cancer (SCLC)

**DOI:** 10.3390/cancers11091297

**Published:** 2019-09-03

**Authors:** Alexander Y. Deneka, Yanis Boumber, Tim Beck, Erica A. Golemis

**Affiliations:** 1Molecular Therapeutics Program, Fox Chase Cancer Center, Philadelphia, PA 19111, USA; 2Department of Biochemistry, Kazan Federal University, 420000 Kazan, Russia; 3Department of Hematology/Oncology, Fox Chase Cancer Center, Philadelphia, PA 19111, USA; 4Cleveland Clinic, Cleveland, OH 44195, USA

**Keywords:** small cell lung cancer, antibody-drug conjugates, targeted therapy

## Abstract

There are few effective therapies for small cell lung cancer (SCLC), a highly aggressive disease representing 15% of total lung cancers. With median survival <2 years, SCLC is one of the most lethal cancers. At present, chemotherapies and radiation therapy are commonly used for SCLC management. Few protein-targeted therapies have shown efficacy in improving overall survival; immune checkpoint inhibitors (ICIs) are promising agents, but many SCLC tumors do not express ICI targets such as PD-L1. This article presents an alternative approach to the treatment of SCLC: the use of drug conjugates, where a targeting moiety concentrates otherwise toxic agents in the vicinity of tumors, maximizing the differential between tumor killing and the cytotoxicity of normal tissues. Several tumor-targeted drug conjugate delivery systems exist and are currently being actively tested in the setting of SCLC. These include antibody-drug conjugates (ADCs), radioimmunoconjugates (RICs), small molecule-drug conjugates (SMDCs), and polymer-drug conjugates (PDCs). We summarize the basis of action for these targeting compounds, discussing principles of construction and providing examples of effective versus ineffective compounds, as established by preclinical and clinical testing. Such agents may offer new therapeutic options for the clinical management of this challenging disease in the future.

## 1. Introduction

An estimated 606,880 Americans will die from cancer in 2019, with lung cancers causing the greatest number of deaths [1]. Although small cell lung cancer (SCLC) only accounts for about 10–15% of lung cancer cases overall, it is responsible for 28,000 deaths annually, making it the seventh most common source of cancer mortality. SCLC is associated with poor survival: post-diagnosis median survival is 15–20 months for patients diagnosed with limited stage (LS) disease (defined as a cancer that is limited to one side of the thorax and adjacent lymph nodes, and treatable by radiotherapy), and 9–12 months for patients with more widely disseminated extensive-stage (ES) disease [2]. The current standard of treatment for LS-SCLC is concurrent radiotherapy and chemotherapy with etoposide in combination with a platinating agent such as carboplatin or cisplatin (etoposide-platinating agent (EP)). However, ~75% of SCLC patients are diagnosed at an advanced stage, when treatment options are limited, treatment resistance rapidly emerges, and dose-limiting toxicities are a major issue [3]. 

Until relatively recently, the first-line treatment for ES patients remained chemotherapy with an EP doublet [4]. Some trials have explored alternative chemotherapy regimens. For instance, in a Japanese randomized phase 3 clinical trial, irinotecan/cisplatin demonstrated superior efficacy with over three months’ improvement in overall survival compared to EP in Japan. However, phase 3 trials conducted in the EU and USA failed to show superiority [5]. Second-line therapies in common use for ES SCLC are generally based on conventional chemotherapies [4], but their efficacy is limited. 

The emergence of immune checkpoint inhibitors (ICIs) in the last decade has attracted considerable interest in SCLC and in other cancers; there is ongoing assessment of numerous ICIs in clinical trials. Immunotherapy with pembrolizumab, a humanized immunoglobulin (Ig) G4 (IgG4) antibody targeting programmed death 1 (PD-1) receptors, demonstrated a durable response (overall response rate (ORR) 33%) in PD-L1-expressing SCLC tumors, even in patients with pretreated SCLC [6]. Nivolumab, a fully humanized IgG4 antibody targeting PD-1, showed a striking response and encouraging improvements in survival in patients with pretreated SCLC, whether used as a monotherapy or in combination with ipilimumab (a fully humanized IgG1 antibody targeting cytotoxic T lymphocyte antigen-4 (CTLA-4)) [7,8]. Recently, FDA approval was granted for the programmed death ligand 1 (PD-L1)-targeting monoclonal antibody atezolizumab as a new standard of targeted therapy in SCLC [9]. Addition of atezolizumab to chemotherapy in the first-line treatment of ES-SCLC resulted in gains in overall survival (OS) and progression-free survival (PFS) versus chemotherapy alone, and quickly became a new treatment standard. However, although these gains were statistically significant, the magnitude of OS improvement was modest (12.3 vs., 10.3 months [9]). These results have led to continued investigations into immune oncology, particularly for ES-SCLC. In SCLC (as in other cancers), higher tumor mutational burden is a direct, independent predictive biomarker for positive response to ICIs [10,11]. Typically, SCLC is characterized by high tumor mutational burden (TMB) [12,13], which is promising for the use of ICIs. However, while other cancers, including non-small-cell lung cancer (NSCLC) [14], have seen transformative effects of ICIs, some factors potentially limit their promise in SCLC. For example, the expression of PD-L1 is relatively uncommon in SCLC (∼28.6% [6]), limiting the pool of tumors likely to respond. 

There are currently few effective protein-targeted therapies for SCLC. In contrast to other cancers, where mutations often occur in targetable signaling pathways (such as kinase-activating mutations [13]), the high TMB of SCLC is typically associated with nontargetable changes. Rather, characteristic features arising from mutations include inactivation of the tumor suppressors TP53 (~85% of cases) and RB1 (~57% of cases), reduction in activity of NOTCH (~14%), altered function of the TP53-related protein TP73 (~13%), and amplification causing overexpression of MYC (~7%) [12]. Nevertheless, there are currently a number of preclinical studies and clinical trials assessing the efficacy of novel targeted therapies for SCLC [15]. Some examples of these include targeting signaling proteins displayed on the cell surface, such as NOTCH [16], the NOTCH ligand DLL3 [17], CD56 (also known as NCAM) [18,19], CD24 [20], CD47 [21], and the proto-oncogene c-KIT (also known as CD117) [22], using inhibitory antibodies or small molecules. There are also ongoing trials involving small molecule inhibitors of PARP (poly-(ADP)-ribose polymerase) [23,24,25] and EZH2 (enhancer of zeste homolog 2), which regulate DNA damage responses and modulate chromatin [26]; and inhibitors of angiogenesis, using various strategies to target VEGF, FGFR1, and CD13 [27,28]. The success of these agents remains to be established.

There remains an urgent need for new therapeutic approaches for the management of SCLC: ideally, approaches that leverage and improve those approaches that have been successful. One critical barrier to the effective treatment of SCLC (and other tumors) with chemotherapies and other cytotoxic drugs is the inability to concentrate these compounds in tumors at sufficient levels to achieve therapeutic benefit without simultaneously inducing untenable degrees of damage to normal tissues. Towards this end, extensive efforts have been devoted to design site-selective drug delivery strategies to deliver chemotherapeutic agents to the tumors. These include antibody-drug conjugates (ADCs) [29,30,31], radioimmunoconjugates (RICs) [32,33], small molecule-drug conjugates (SMDCs) [34], and drug conjugates to tumor-targeting peptides and synthetic polymers (PDCs) [35,36]. Such conjugate-based drug delivery systems share commonalities with respect to their functional construction, comprising a tumor recognition moiety and cytotoxic payload connected via chemical linkage. Some examples of this approach have been proven to be successful in some solid tumor settings, as demonstrated by the recent FDA approval of the antibody‒drug conjugates brentuximab vedotin (Adcetris) in Hodgkin’s lymphoma [37], ado-trastuzumab emtansine (Kadcyla) in HER2+ breast cancer [38,39], and polatuzumab vedotin-piiq (Polivy) in relapsed or refractory diffuse large B-cell lymphoma [40,41]. In this review, we summarize the relative promise of the numerous tumor-targeted drug conjugate therapies currently under evaluation in preclinical studies and in clinical trials for SCLC.

## 2. Antibody-Drug Conjugates (ADCs) and Radioimmunoconjugates (RICs)

The German physician Paul Ehrlich first proposed the concept of selectively delivering a cytotoxic drug to a tumor via a targeting agent at the beginning of the 20th century [42]. It took some decades to introduce this concept to practice, and the efficient production of monoclonal antibodies of defined ligand specificity provided a critical enabling step for widespread application. However, ADCs and RICs exploit Ehrlich’s original concept to target tumors. Compared with conventional chemotherapies, these agents extend the therapeutic window, lowering the minimum effective dose for tumors and increasing the maximum tolerated dose because antibody-mediated drug delivery causes fewer systemic effects [43]. 

### 2.1. ADCs

ADCs have a tripartite structure, including an antibody or antibody fragment, a linker, and a cytotoxic moiety (Figure 1). Initially, the antibody component was typically derived from a mouse monoclonal antibody (mAb); in recent years, humanized antibodies have resulted in a significant improvement of therapeutic properties [44,45]. When targeting antibodies are conjugated with cytotoxic payloads, they typically retain any original ability to activate immune responses such as antibody-dependent cellular cytotoxicity (ADCC) or complement-dependent cytotoxicity [45], augmenting tumor-killing activity.

For optimal ADC function, the antibody moiety targets an antigen that is highly and homogeneously expressed on tumors to be targeted, expressed at low levels or absent on normal tissue, and expressed in the extracellular space of tumor cells but not shed from the tumor surface (to prevent unspecific binding within the circulation) [46]. Human epidermal growth factor receptor (EGFR) and related proteins epidermal growth factor receptors 2 and 3 (HER2, HER3), prostate-specific membrane antigen (PSMA), and c-MET are examples of successfully targeted antigens expressed on both normal and tumor tissues, although typically at higher levels on tumors [47]. However, some studies indicate that targeting antigens with relatively low expression in tumors is feasible, particularly if there is minimal expression in normal tissue [45]. In cases where there is not homogeneous expression of the target antigen in solid tumors, this issue is sometimes rendered moot by the efficient bystander effect: a process by which an antigen-positive tumor cell internalizes the ADC and cleaves the cytotoxic payload from the linker, releasing a membrane-permeable moiety that can induce cell death in neighboring tumor cells that lack the antigen [45,48]. However, this is not ideal, as such cleavage and release may result in an increased rate of off-target systemic toxicity. 

Other features of effective ADCs include antibody moieties that are efficiently internalized by receptor-mediated endocytosis of the target antigen, and those for which that expression of the target antigen is not downregulated after treatment of a tumor with the ADC [48]. Other factors such as tumor size and properties of intratumoral cellular organization, including degree of fibrosis, tumor vascularization, and other physical barriers, can diminish payload uptake [45,48]. Some studies have shown that targeting not only tumor cells, but also components of the tumor microenvironment such as the stromal fibroblasts, the tumor vasculature [49,50], and tumor-initiating cancer stem cells [49,51], is an alternative way to target tumors with ADCs. 

The linker moiety plays a crucial role in ADC pharmacokinetics, pharmacodynamics, and efficacy [44,46,52]. Ongoing research seeks to optimize stable linkers that prevent ADCs from release of the cytotoxic drug before reaching its tumoral target, to prevent off-target toxicity, but which allow effective release of the payload once the ADC is internalized by the target tumor cell [53]. Examples of linkers that have been successfully deployed include both noncleavable and cleavable linkers. Noncleavable linkers, such as are used in ado-trastuzumab emtansine, rely on complete lysosomal degradation of the antibody moiety to allow the cytotoxic payload to act [46]. Cleavable linkers are designed to release payload molecules after bond cleavage. Such linkers can be acid-sensitive, so that they release both the free drug and the antibody selectively in the low-pH conditions present in lysosomes or endosomes; this approach is used in the ADCs gemtuzumab, ozogamicin, and inotuzumab ozogamicin [46]. Other types of cleavable linker are sensitive to proteases found specifically in the lysosome (an approach used in the ADC brentuximab vedotin) [54], or require glutathione as a cofactor for cleavage, where targeting benefits from the higher concentration of glutathione in tumor cells [51].

The cytotoxic payload is the third major component of an ADC. Optimally, payloads should be soluble, amenable to conjugation, and stable. They typically have extremely high cytotoxic potency for intracellular targets. The first generation of ADCs used classical chemotherapy agents, including doxorubicin and methotrexate, as payloads [46]. Payloads currently undergoing assessment in clinical trials generally fall into three categories: antimitotic (often tubulin filament-damaging), DNA-damaging, and transcriptional inhibitors. 

Tubulin filament-targeting agents that disrupt mitosis include maytansinoids, which suppress microtubule dynamic instability, inducing mitotic arrest. This is similar to the mode of action of vinblastine [55] and taxol derivatives, which promote microtubule assembly and inhibit disassembly via direct interaction with microtubules, which enhances tubulin polymerization, causing G2/M arrest [56]. The maytansinoid mertanzine (DM1) is a part of the FDA-approved ADC ado-trastuzumab emtansine [38]. Other tubulin-targeting payloads include auristatins, which bind α,β-tubulin during interphase to inhibit tubulin-dependent guanosine triphosphate (GTP) hydrolysis and microtubule assembly [51,57]. The payload monomethyl auristatin E (MMAE) is successfully used in the treatment of Hodgkin lymphoma as a part of the ADC brentuximab vedotin [37]. Tubulysin analogs fall into a third group of antimitotic agents; these cause rapid disintegration of the cytoskeleton and mitotic machinery of dividing cells, leading to apoptosis [58]. 

Among DNA-damaging payloads, duocarmycin is a powerful alkylating compound that demonstrated potency in multidrug-resistant cancer models. Duocarmycin-based ADCs conjugated to the anti-HER2 antibody trastuzumab underwent phase 1 investigation in clinical trials, with trial results pending (NCT02277717, see Table 1) [59]. Calicheamicin causes double-strand DNA breaks in a manner independent of the cell cycle stage, making it useful against tumors with lower rates of proliferation [51,60]. The FDA has approved two ADCs bearing calicheamicin as a payload—gemtuzumab ozogamicin in acute myeloid leukemia (AML), and inotuzumab ozogamicin in acute lymphoblastic lymphoma (ALL) [61,62]. The topoisomerase inhibitor SN-38 (the active metabolite of irinotecan) has been used in numerous trials as an antibody conjugate, with an ongoing trial in SCLC (NCT01631552, discussed further below) [63]. Pyrrolobenzodiazepines (PBDs) bind to discrete DNA sequences, causing multiple DNA cross-links, and have attracted interest because they do not present any cross-resistance with common chemotherapeutic agents [64]. PBDs conjugated with an antibody to c-MET are currently being assessed in a clinical trial for c-MET-positive solid tumors, NCT03859752 [57]. Alpha-amanitin is a potent RNA polymerase II inhibitor, derived from the mushroom *Amanita phalloides*; this agent is showing promising results in preclinical assessment in a pancreatic cancer setting, when used as part of an ADC with antibodies targeting epithelial cell adhesion molecules (EpCAMs) [65].

### 2.2. Clinical Use of ADCs for SCLC

As of the middle of 2019, three ADCs have been licensed by the FDA for solid tumor malignancies. Ado-trastuzumab emtansine (T-DM1, Kadcyla) [38,39] has been assessed in clinical trials in NSCLC, where it demonstrated a 44% partial response rate and a five-month PFS in HER2-positive patients [66]. Brentuximab vedotin (Adcetris) [37] and polatuzumab vedotin-piiq (Polivy) [40,41], also licensed, have not yet been investigated in lung cancer. 

T-DM1 is composed of trastuzumab, a humanized IgG1 anti-HER2 antibody fused via a noncleavable linker to the maytansinoid DM1 [38]. T-DM1 has a tolerable toxicity profile [38] and merits potential investigation in SCLC clinical trials, as preclinical assessment of this compound in vivo in trastuzumab-resistant HER2-positive SCLC xenografts demonstrated that-DM1 was able to induce apoptosis in SBC-3/SN-38 xenografts to a greater extent than trastuzumab monotherapy [67]. Brentuximab vedotin (BV, Adcetris) is composed of an anti-CD30 mAb connected via a cleavable peptide to MMAE [68], while polatuzumab vedotin-piiq (Polivy) is composed of B-cell antigen receptor complex-associated protein beta chain (CD79B) conjugated via a protease-cleavable peptide linker, to MMAE [41]. Although highly potent against Hodgkin’s lymphoma, where the CD30 ligand is highly expressed [37,54], and non-Hodgkin’s lymphoma, which abundantly expresses CD79B [69], these two ADCs are less immediately promising for SCLC, where these antigens are not known to be expressed. Overall, research addressing the clinical utility of ADCs is a very active field, with almost 120 ongoing clinical trials involving over 60 unique ADCs. Although most of these trials focus on hematological malignancies, some show promising results in the setting of SCLC or other lung cancers (Table 1). 

Among the agents under investigation for SCLC is rovalpituzumab tesirine (SC16LD6.5 or Rova-T): the antibody moiety (SC16) targets DLL3, a surface marker of tumor-initiating cells, expressed in about 80% of SCLC patients. SC16 is conjugated to D6.5, a small molecule that is a potent, cell cycle-independent DNA-damaging agent. A phase 1 trial of this agent (NCT01901653) demonstrated encouraging single-agent antitumor activity with a manageable safety profile [70], leading to several additional trials that investigated Rova-T as a frontline and second-line treatment in ES-SCLC (NCT0281999, NCT03334487). Unfortunately, despite encouraging interim data reported in press releases in 2018, the phase 3 second-line trial was terminated due to Rova-T arm patients having inferior survival compared to patients on topotecan [71]. However, Rova-T has not yet been abandoned as a potentially useful clinical agent for SCLC. The results of a study investigating Rova-T as a third-line treatment in 199 patients with ES-SCLC with and without DLL3 expression (NCT02674568) were published in 2018 [72]. This study demonstrated positive results, with an overall response rate (ORR) of 38% and progression-free survival (PFS) of 4.3 months in DLL3-positive patients, but no benefit in an unselected cohort, which had an ORR of 18% and PFS of 2.8 months. Currently, Rova-T is being assessed as a maintenance therapy in ES-SCLC following first-line platinum-based chemotherapy (NCT03033511). 

CD56/NCAM is an attractive target for SCLC, as it is expressed in almost all SCLC tumors [73]. In lorvotuzumab mertansine, the humanized anti-CD56 antibody lorvotuzumab is joined via a cleavable disulfide linker to the maytansinoid DM1. This ADC demonstrated significant antitumor potency in preclinical SCLC human xenograft studies, both as a monotherapy and in combination with chemotherapy (carboplatin/etoposide) [19]. However, a phase 1/2 clinical trial with this compound in ES-SCLC was terminated due to a lack of efficacy and safety concerns (NCT01237678) [18]. Currently, this compound is undergoing phase 2 trial for neuroblastoma, Wilms tumors, and several other malignancies in younger patients (NCT02452554); it is possible that it may find use as a combination agent for SCLC. Another anti-CD56 antibody conjugate that demonstrated encouraging results in preclinical assessment is promiximab-duocarmycin, in which a CD56 antibody is conjugated to a potent DNA alkylating agent, duocarmycin (DUBA) using a novel linker with reduced interchain disulfides. This ADC demonstrated high efficacy in SCLC xenograft models [74], suggesting that further clinical assessment of this compound may be useful.

The ADC sacituzumab govitecan consists of a humanized mAb targeting Trop-2 (trophoblastic antigen-2), which is highly expressed in numerous epithelial cancers [75], fused to SN-38 (the active metabolite of irinotecan, [76]), which inhibits topoisomerase I to induce double- and single-strand DNA breaks. This conjugate demonstrated an encouraging result in a phase 1 clinical trial, with a predictable pharmacokinetic profile and manageable toxicity [77]. A phase 1/2 trial (NCT01631552) assessed its activity in patients with advanced cancers, where it demonstrated positive results in heavily pretreated hormone receptor (ER+/PR+)-positive/HER2-negative metastatic breast cancer patients, and was well tolerated, with a safety profile consistent with the reports of the previous trial [78]. Evaluation of sacituzumab govitecan in a single-arm study performed in heavily pretreated metastatic SCLC patients, including patients who were chemoresistant or chemosensitive to first-line chemotherapy, indicated some efficacy, with an overall response rate of 14% and a 34% clinical benefit rate (in this study defined as complete response + partial response + stable disease ≥ 4 months). The median OS was 7.5 months, and the median PFS was 3.7 months [79]. Currently sacituzumab govitecan is undergoing several trials, including one phase 1/2 trial in SCLC, where it is being evaluated as a single agent in previously treated patients with advanced SCLC or other epithelial cancers (NCT01631552). 

c-KIT overexpression is well recognized in numerous malignancies, including SCLC [80], as well as melanoma [81], non-small-cell lung cancer (NSCLC) [82], acute myelogenous leukemia (AML) [83], and gastrointestinal stromal tumors (GIST) [84]. LOP628 comprises a humanized anti-c-KIT antibody LMJ729, conjugated via a noncleavable linker to DM1; this ADC exhibited potent antiproliferative activity in c-KIT-positive cell lines versus antibody alone in cell culture and murine xenograft models [22]. The positive results of this preclinical study strongly suggest further clinical evaluation of LOP628 in patients with SCLC, GIST, and AML. 

Tucotuzumab celmoleukin is a fusion protein developed along the concepts of an ADC [85]. This agent is comprised of the humanized mAb tucotuzumab (which recognizes EpCAM, commonly expressed on SCLC tumors [86]) fused to active interleukin-2 (IL2). Subsequently, the localized IL2 moiety of this fusion protein helps stimulate a local cytotoxic T-cell antitumor immune response. Clinical assessment of this compound (NCT00016237) in 64 patients with ES-SCLC in combination with cyclophosphamide demonstrated a trend toward prolonged PFS and OS versus supportive care (*n* = 44), particularly in a subgroup of patients receiving previous prophylactic cranial irradiation (*n* = 26) [85]. These data suggest the potential for statistically significant results in a larger population.

### 2.3. RICs

Designed in line with the principles of ADC construction, in radioimmunoconjugates (RICs), radionuclides are used as the payloads linked to tumor-targeting antibodies (Figure 1). The use of RICs has several specific limitations, because of the nature of their radioactive payloads. There is a need for a reliable supply chain of radionuclides and the production pipeline can be expensive. The delivered dose of radionuclides is dependent on the antibody component for pharmacokinetic biodistribution, and hence has a low dose rate at the tumor site approximately two orders of magnitude lower in intensity compared with conventional external beam radiation therapy, making it difficult to measure the dose–response relationship with outcomes in patients [87,88]. However, RICs also have some advantages, including their use in tumor imaging.

Current applications for RICs include diagnostic immunoscintigraphy and radioimmunotherapy [33]. For diagnostic applications, the preferred radionuclides are positron (β+)-emitting isotopes, such as short half-life ^68^Ga and ^44^Sc, and long half-life ^64^Cu and ^89^Zr, which enable high-resolution positron emission tomography (PET) imaging [89,90]. (β−)-emitting and α-emitting isotopes are used in therapeutic applications. Radionuclide choice is based on the tumor size: ^90^Y exhibits long-range (β) emission and can be used for larger masses; ^177^Lu and ^188^Re have a short range, favoring treatment of minimal, residual disease [33]. α-emitting isotopes, such as ^225^Ac, ^213^Bi, and ^211^At, deliver a high proportion of their energy inside the targeted cells, leading to highly efficient killing [91,92]. The current consensus of treatment is that RICs may be most useful for cases of small disseminated tumors, clusters of malignant cells, or residual disease, particularly in cases where the antibody component can assure significant tumor-specific targeting. Currently, RICs are under investigation in a number of solid tumors, particularly in treating minimal residual disease in prostate and colorectal cancer [33], and are also being explored for SCLC [93,94]. 

### 2.4. RICs’ Clinical Use in SCLC 

The first clinical application of radioimmunotherapy was for the treatment of non-Hodgkin’s lymphoma [95]. Two RICs targeting CD20 have been approved for the treatment of this disease: ^90^Y-ibritumomab tiuxetan, and ^131^I-tositumomab [96]. In this study ^90^Y-ibritumomab demonstrated a 77.8% response rate (RR) with a 41.7% complete response (CR) in a group of 36 patients, and ^131^I-tositumomab demonstrated similar numbers (70.9% RR and 35.5% CR); however, despite its efficacy, tositumomab production was later discontinued for market reasons, given the availability of other effective treatments for this disease. RICs targeting CD20 are potentially of interest in SCLC: as in lymphomas, CD20 is widely expressed in SCLC tumors [97], and was demonstrated to be associated with clinical prognosis for SCLC [98]. Another promising target for RICs in SCLC is somatostatin receptor (SSTR), which is overexpressed in ~48% of SCLC cases [98]. A phase 1 trial assessing the efficacy of ^188^Re-P2045, a β-emitter conjugated to a somatostatin analog peptide, in both NSCLC and SCLC demonstrated good tolerability of the compound and stable disease [99]. Besides justifying further exploration of this RIC, these data suggest that a similar SSTR2 targeting approach with other compounds may be useful in SCLC.

DLL3 expression can serve as an immunoscintigraphy imaging biomarker for SCLC [93]. Recently, a RIC in which ^89^Zr-labeled SC16 (a mAb targeting DLL3) was designed as a companion diagnostic agent to facilitate the selection of patients for treatment with rovalpituzumab teserine (Rova-T) based on a noninvasive interrogation of the in vivo status of DLL3 expression using PET imaging [94]. In this study, DLL3-guided immunoPET yielded a rank-order correlation for response to Rova-T therapy in SCLC patient-derived xenograft models. At present, the development of RIC compounds is mainly ongoing in the public, academic sector. However, the pharmaceutical industry is beginning to focus more on this technology as promising data emerge [89,90,100,101].

## 3. Small Molecule-Drug Conjugates (SMDCs)

Small molecule-drug conjugates (SMDCs) are designed along similar principles as ADCs for drug delivery and tumor targeting applications, with the difference being that the antibody component is replaced by a targeting ligand that can be a peptide or a small molecule (Figure 2) [102,103]. SMDCs have a number of strengths compared to ADCs. They are frequently easier to synthesize than biological agents. Most are nonimmunogenic, making them unlikely to provoke an autoimmune response [104,105]. Transportation, storage, and administration are easier than with ADCs [106]. Molecular weights of SMDCs are much lower then those of ADCs, resulting in better cell permeability (particularly in solid tumors that may be poorly vascularized) [107]. The low molecular weight and other chemical features are also associated with better in vitro and in vivo stability than biological agents including mAbs [108]. Notably, SMDCs are more rapidly removed from the blood through glomerular filtration in the kidneys than are ADCs. This results in a better toxicity profile; however, it also has the potential to reduce the effective time on the tumor target [109].

The targeting ligand of an SMDC typically has a binding affinity within the nanomolar range for its primary target [110], and potent target selectivity. These properties also help to decrease systemic toxicities upon compound administration [103]. Typically, targeting moieties have been based on well-defined inhibitors or ligands of transmembrane receptors or other enzymes relevant to signaling pathways that are highly active in, and often selective for, tumor cells. SMDCs have used imatinib (a BCR/ABL fusion protein inhibitor, [111]), folic acid (a ligand of the folate receptor, [112]), glucose transporter 1 (GLUT1) [113], low-density lipoprotein receptor-related protein 1 (LRP1) [114], prostate-specific membrane antigen (PSMA) [115], aminopeptidase N (APN) [116], somatostatin receptor (SSTR) [103], and inhibitors of heat shock protein 90 (HSP90) [117,118]. 

The linker in SMDCs, which consists of a spacer and cleavage bridge, plays a role similar to the role of the linker component in ADCs. Linkers are designed to preserve the activity of post-cleavage species and to optimize the drug release, pharmacokinetics, and pharmacodynamics of the targeting ligand and payload [103,112]. First-generation spacers contain carbohydrate units, repeating acidic residues, and saccharic acid residues. Second-generation spacers use glutamic acid and glutamine as epimerization-inert modules [119]. The cleavable bridge retains stability during the SMDC transportation from the vasculature to the tumor, and is typically cleaved by one of two triggering methods. The first mechanism is cleavage in the endosomes of the target cells due to low pH. Such a cleavage bridge comprises acetals and hydrazones [120]. The second mechanism is through use of a disulfide-based linker, which undergoes cleavage due to an intracellular excess of glutathione (GSH), thioredoxin, peroxiredoxins, and nicotinamide adenine dinucleotides (NADH and NADPH) [103]. 

The final component of SMDCs is the small molecule payload. As with ADCs, optimal payloads have high binding affinity for their targets [103] and are highly cytotoxic, similar to those used in ADCs. In some cases, to increase the cytotoxic activity of the conjugate, multivalent ligands, comprising several payloads linked to the targeting compound, are employed. Examples of payloads that target mitosis, DNA replication, and protein translation that are currently in assessment as SMDCs in lung cancer and other settings are listed in Table 1.

### Clinical Perspective on the Use of SMDCs in SCLC

Among the SMDCS under preclinical and clinical development, several recent preclinical studies demonstrated the striking potency of HSP90 inhibitor drug conjugates such as PEN-866 (formerly STA-8666) in xenograft models of solid tumors, including breast, pancreatic, and SCLC [121,122,123]. As a result of stresses existing within tumors and the tumor microenvironment, HSP90 is highly overexpressed in tumors relative to normal tissue [124,125]. PEN-866/STA-8666 is comprised of an HSP90-targeting moiety fused via a cleavable carbamate linker to the cytotoxic compound SN-38, an active metabolite of irenotecan. In a set of preclinical studies, this SMDC exhibited extended intratumoral drug exposure and superior therapeutic indices over irinotecan therapy due to its concentration in tumor tissue, where intratumoral cleavage provides high, selective SN-38 exposure for up to a week [121,122,123,126]. Benchmarking of STA-8666/PEN-866 against standard first- and second-line therapies showed superior performance, and exceptional potency, even in SCLC patient-derived xenografts (PDX) and SCLC cell line models previously exposed to other chemotherapies, including irenotecan [122]. Coupled with the identification of response biomarkers, these studies provided significant preclinical support for the development of this agent towards the clinic. Currently, PEN-866 is being evaluated for efficacy in solid tumors, including SCLC, in a phase 1/2 trial (NCT03221400).

Folic acid (folate) conjugates comprise another class of SMDCs assessed in numerous cancer settings, including lung cancer. Although they have not yet been specifically assessed in SCLC, folate is likely to be a good targeting agent in this cancer because recent studies demonstrate that folate receptor-positive circulating tumor cells can serve as a valuable predictive and prognostic biomarker for patients with SCLC who received first-line chemotherapy [127]. The promise of these agents was demonstrated in phase 1 studies in folate receptor positive refractory solid tumors, and subsequent phase 2 studies in ovarian cancers [128,129], suggesting that they may be useful tools for SCLC. SN-38 has also been expressed as an SMDC with folic acid using a hydrophilic peptide spacer and a releasable disulfide carbonate linker. This conjugate exhibited high affinity for folate receptor-expressing cells, and inhibited the proliferation of folate receptor-positive human cervical cancer KB cells with an IC50 within a nanomolar range [130]. 

In folate-vinca alkaloid SMDCs, the payload is a microtubule-destabilizing agent that inhibits the assembly of microtubules and causes defects in mitosis. The SMDC drug delivery strategy enhances selectivity and reduces the toxicity of a group of agents, the vinca alkaloids, that are independently active in many cancers. A series of folate conjugates to various vinca alkaloids (including vinorelbine, vindesine, and others) using a common cleavable linker has been extensively studied. Promising conjugates identified in this study include vintafolide (EC145) and its analogs [131,132,133]. In a phase IIb study in NSCLC of EC145 used alone and in combination with docetaxel completed in 2015 (NCT01577654), the combination improved the PFS to 7.1 months and the overall survival to 10.9 months, reducing the risk of disease worsening or death by 25% (PFS HR 0.75, *p* = 0.0696, one-sided test). Based on these initial successes, folate-vinca alkaloid conjugates with second-generation spacers were developed (EC0489, EC0492). These compounds have further reduced toxicities compared to first-generation compounds (70% less than EC145) and better elimination, as demonstrated in murine preclinical studies [131]. A phase I study (NCT00852189) of EC0489 for the treatment of refractory and metastatic tumors in patients who have exhausted the standard treatment options has been completed, demonstrating that patients can receive doses of EC0489 equivalent to twice the amount of EC145. 

The folic acid biconjugate SMDC EC0225 (Novartis) consists of two drugs—desacetylvinblastine monohydrazide (DAVLBH), a derivative of vinblastine, and mitomycin C (a potent alkylating agent)—tethered to a single folate unit. This conjugate demonstrated high potency and specificity against folate receptor-positive nasopharyngeal, NSCLC, and breast tumors [134]. EC0225 is currently undergoing phase I clinical trial assessment (NCT00441870) for patients with refractory or metastatic tumors. 

Other microtubule-targeting agents are being investigated as payloads in folate-targeted SMDCs. For example, folate-taxol conjugates are being assessed in preclinical studies. Folic acid-5-aminofluorescein-glutamic-paclitaxel demonstrated improved water solubility, loading rate, targeting ability, and antitumor activity, and toxicity profiles compared to paclitaxel alone in breast, lung, and kidney cancer mouse models [135]. Tubulysins are also being assessed in preclinical studies as folic acid conjugates (EC0305) [131], and demonstrated more potent antitumor activity than folate-vinca SMDC EC145 (see above) in two distinct drug-resistant, folate-receptor-expressing tumor cell lines M109 and 4T1-cl2, both in vitro and in vivo [136]. Phase 1 evaluation of conjugates of folic acid and tubulysin with second-generation spacers (EC1456 and EC20) in advanced solid cancers (NCT01999738) has just been completed, demonstrating good tolerability and efficacy, as suggested by durable stable disease [137]; more detailed study results are pending. 

Phosphatidylserines (PS) are phospholipid components of cell membranes that in normal cells are tightly regulated to be asymmetrically segregated in the inner leaflet of the cell [138]. However, PS are expressed on the surface of numerous tumor cell lines and solid tumors, as well as in tumor vasculatures [139,140]. A study by Sanchez-Rodriguez et al. has suggested that PSs can serve as a biomarker of SCLC, indicating their relevance to this tumor type [141]. Several studies demonstrated that radiation results in an up to 5-fold increase in the expression of surface PSs on tumor blood vessels, and this has been successfully exploited in preclinical studies in murine models of lung cancer with the PS-targeting antibody bavituximab [142,143], and in clinical trials in numerous solid cancer malignancies, including NSCLC [144,145,146], indicating that PS are useful for tumor targeting in the lungs. Small molecule zinc (II) dipicolylamine (ZnDPA) was initially described as an imaging probe selectively targeting the surfaces of anionic PS-containing cell membranes [147]. A recent preclinical study described another promising example of an SMDC based on a ZnDPA-SN-38 conjugate that has effective PS-targeted delivery properties and a good toxicity profile both in vitro and in vivo. This work suggests the potential use of SMDCs agents conjugated in the same manner to treat PS-associated malignancies [148].

For all the SMDCs discussed above, conjugation to a targeting moiety led to higher potency, specificity, and better safety profiles due to greater tumor versus normal tissue accumulation than the payload administered alone. This efficacy, coupled with the easier synthesis and administration relative to ADCs, suggests that SMDCs are a particularly promising approach for the treatment of SCLC.

## 4. Polymer Drug Conjugates and BiTE Antibodies for ES-SCLC

Several additional strategies for the targeted delivery of cytotoxic compounds are currently under investigation in preclinical and clinical studies; although they have not yet been investigated extensively in SCLC, they are briefly noted here. 

Polymer-drug conjugates (PDCs) seek to optimize the targeted delivery of cytotoxic payloads to the tumors based on an improvement in drug availability. Several compounds have been approved by the FDA for use in solid tumors. Doxil is a 100 nm PEGylated liposome conjugated with doxorubicin for the treatment of breast cancer, ovarian cancer, and other solid tumors [149]. Abraxane is a 130 nm albumin-stabilized paclitaxel nanoparticle, used for the treatment of metastatic breast cancer [150,151]. Onivyde is a nanoliposome conjugated with irinotecan for pancreatic cancer management [152]. These PDCs typically demonstrate better pharmacokinetics and reduced adverse effects compared to their payloads alone, because of the enhanced cancer cell permeability and tumor retention achieved due to the physical and chemical properties of the nanoparticles. Tumor-targeting is also attained through the incorporation into the polymeric component of PDCs of elements that respond to triggers (such as tumor pH, which is typically lower, or tumor-concentrated enzymes), or a locally applied stimulus (light or heat) that can be externally applied to selectively mark tumors [153]. The polymer-conjugated targeting approach has not yet been extensively investigated in the SCLC setting; however, several studies support the likely efficacy of this method. First, nanoparticle delivery of *TP53* conjugated with poly (β-amino esters) into SCLC cells, demonstrated activity in controlling cancer cell growth both in vitro and in vivo [154], implying efficient uptake into tumor tissues. As another example, CRLX101 is a novel cyclodextrin-containing polymer conjugate of camptothecin that self-assembles into nanoparticles and delivers sustained levels of active cytotoxic moiety into cancer cells while substantially reducing systemic exposure [155]. Based on promising preclinical data, CRLX101 is currently being evaluated in a series of phase 1/2 clinical trials, including in the SCLC setting (NCT02769962), both as a monotherapy and in combination with the PARP inhibitor olaparib.

A relatively new strategy that is being explored and that is conceptually related to the use of drug conjugates is the use of bispecific T cell engager (BiTE) antibody constructs, as an alternative to ICIs [17]. For example, the BiTE AMG757 is designed to transiently connect DLL3-positive SCLC tumor cells to CD3-positive T cells and induce T cell-mediated cell lysis and concomitant T cell proliferation. It achieves this by incorporating two single-chain variable fragments (scFv), targeted to DLL3 and CD3, and fused via a short, flexible glycine-serine linker [156]. In preclinical studies, AMG757 demonstrated potent killing of SCLC cell lines in vitro and tumor growth suppression in the SHP-77 human SCLC xenograft model in vivo [17]. This agent is currently undergoing phase 1 trial assessment in patients with SCLC (NCT03319940) [157].

## 5. Conclusions

SCLC is a devastating disease, and poses many treatment challenges. Given the frequent diagnosis of this disease at an advanced stage, and in the absence of common response to targeted agents and ICIs, it is essential to optimize the response to classic tools of cancer control: chemotherapies and radiotherapies. Drug conjugates try to address one of the greatest challenges: the inability of potent cytotoxic compounds to demonstrate full antitumor activity because of systemic toxicities emerging from the inability to concentrate the drug within the tumor tissue in high concentrations, while minimizing systemic side effects. The strategies summarized above represent variations on site-selective drug “trojan horse” delivery strategies, designed to effectively deliver potent chemotherapeutic agents towards the SCLC and other tumors. At this point, it is difficult to conclude which will be the most effective, and whether they will be most useful in frontline settings in combination with other approaches (such as ICIs), or whether they are more likely to function as salvage agents. 

For many, but not all, of the agents discussed, biomarkers for their effective use include the protein ligand for their targeting moiety; whether other biomarkers might also be developed has not been well explored. Given that most of the agents include DNA-damaging or antimitotic agents, germline or somatic mutations in genes governing response to such agents may greatly influence the response (for instance, BRCA mutations, or other mutations affecting DNA repair or cell cycle checkpoints [158]). Tumor vascularity is also likely to play a major role in influencing the treatment response with these agents, and may influence patient selection. For those tumors that are diagnosed at an early stage (LS-SCLC) and are amenable to radiation therapy, it is possible that the neoadjuvant or concurrent use of drug conjugates to cytotoxic agents may help in maximizing antitumor response. 

Finally, the combination of ADCs with ICIs is worth evaluation, given the fact that the latter are effective and FDA-approved for a number of cancer types, and since ADCs can independently elicit immune responses [159]. A caveat to this approach is that, at present, most SCLC patients are exposed to first-line ICIs; hence, a clinical trial to probe the value of combining an ADC with an ICI as a second-line treatment, in previously ICI-exposed SCLC patients, may be a useful trial design. Overall, drug conjugates show considerable promise in controlling or eliminating SCLC tumors, and may offer new therapeutic options for the clinical management of this disease in the future.

## Figures and Tables

**Figure 1 cancers-11-01297-f001:**
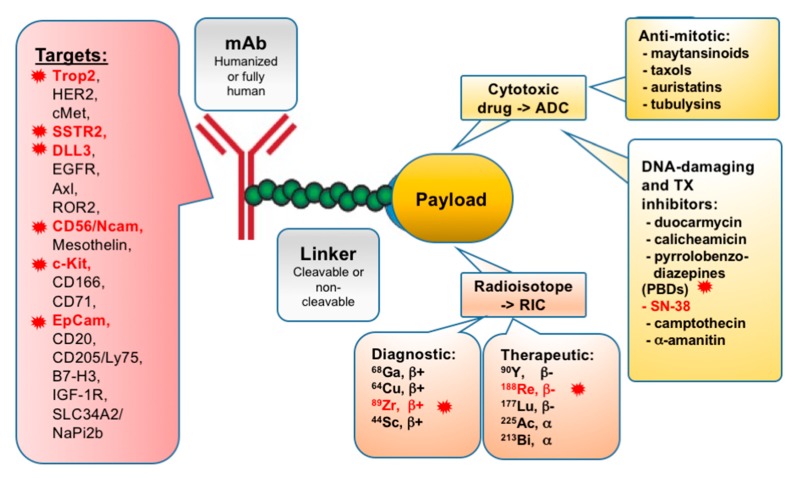
Structure, target antigens in solid tumors, and mechanism of action of antibody-drug conjugates and radioimmunoconjugates. Highlighted in red are the targets, cytotoxic compounds, and radioisotopes assessed in small cell lung cancer (SCLC) trials.

**Figure 2 cancers-11-01297-f002:**
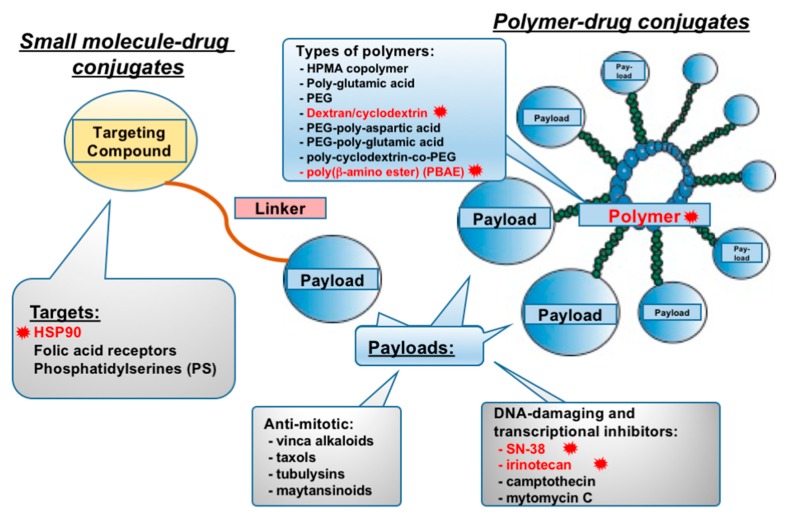
Small molecule-drug conjugates (SMDCs) and polymer-drug conjugates (PDCs). Schematic representation of structural features, targets, and mechanism of action. While sharing common payloads, the targeting mechanism differs (see text). Highlighted in red are the targets, polymers, and cytotoxic compounds assessed in the SCLC setting; black font indicates moieties under investigation and showing promise in other cancers, but not yet in SCLC.

**Table 1 cancers-11-01297-t001:** List of active trials assessing tumor-targeted drug conjugates in lung cancers. ADC, antibody-drug conjugate; RIC, radioimmunoconjugate; PDC, polymer-drug conjugate; SMDC, small molecule-drug conjugate.

Drug Name	Type of Conjugate	Disease	Target	Payload	NCT#	Phase
IMMU-132 (sacituzumab govitecan)	ADC	SCLC, NSCLC, other epithelial cancers	Trop-2	SN-38 (topoisomerase I inhibitor)	NCT01631552	1, 2
Rovalpituzumab tesirine (Rova-T)	ADC	SCLC, solid tumors	DLL3	Pyrrolobenzodiazepine dimer (PBD) (DNA cross-linking)	NCT03033511, NCT03000257	3, 1
A166	ADC	Lung cancer, other HER2+ cancers	HER2	undisclosed	NCT03602079	1, 2
ABBV-399 (telizotuzumab vedotin)	ADC	NSCLC	cMet	Monomethyl auristatin E (MMAE) (antimitotic)	NCT02099058, NCT03539536	1
AVID100	ADC	NSCLC, other solid tumors	EGFR	Maytansinoid mertansine DM1	NCT03094169	1, 2
BA3011 (CAB-AXL)	ADC	NSCLC, other solid tumors	Axl	Monomethyl auristatin E (MMAE) (antimitotic)	NCT03425279	1, 2
BA3021 (CAB-ROR2)	ADC	NSCLC, other solid tumors	ROR2	undisclosed	NCT03504488	1, 2
BAY 94-9343 (anetumab ravtansine)	ADC	NSCLC, mesothelin positive, others	mesothelin	Maytansinoid DM4 (antimitotic)	NCT01439152, NCT03455556	1
BMS-986148	ADC	NSCLC, other solid tumors	mesothelin	Duocarmycin-related (DNA-alkylating agent)	NCT02341625	1, 2
CX-2009	ADC	NSCLC, other solid tumors	CD166	Maytansinoid DM4 (antimitotic)	NCT03149549	1, 2
CX-2029	ADC	NSCLC, other solid tumors	CD71	Monomethylauristatin E (MMAE) (antimitotic)	NCT03543813	1, 2
DS-8201a	ADC	NSCLC, HER2 positive	HER2	Topoisomerase I inhibitor	NCT03505710, NCT02564900	2
Enapotamab vedotin	ADC	NSCLC, other solid tumors	Axl	Monomethylauristatin E (MMAE) (antimitotic)	NCT02988817	1, 2
FS-1502 (Trastuzumab Monomethyl Auristatin F)	ADC	NSCLC, breast and other solid tumors	HER2	Auristatin F-HPA (antimitotic)	NCT03944499	1
MEN1309	ADC	Metastatic NSCLC, other solid tumors	CD205/Ly75	Maytansinoid DM4 (antimitotic)	NCT03403725	1
MGC018	ADC	NSCLC, other solid tumors	B7-H3	Duocarmycin (DNA-alkylating agent)	NCT03729596	1, 2
SHR-A1403	ADC	NSCLC, other solid tumors	cMet	Microtubule inhibitor	NCT03856541	1
SYD985 (trastuzumab vc-seco-DUBA)	ADC	NSCLC, other solid tumors	HER2	Duocarmycin (DNA-alkylating agent)	NCT02277717	1
TR1801	ADC	NSCLC, other solid tumors	cMet	Pyrrolobenzodiazepine dimer (PBD) (DNA cross-linking)	NCT03859752	1
U3 1402	ADC	NSCLC	HER3	Topoisomerase I inhibitor DX 8951	NCT03260491	1
W0101	ADC	NSCLC, other solid tumors	IGF-1R	Auristatin derivative (antimitotic)	NCT03316638	1, 2
XMT-1522	ADC	NSCLC, breast cancer	HER2	Multiple, x15 auristatin molecules (antimitotic)	NCT02952729	1
XMT1536	ADC	NSCLC, ovarian cancer	SLC34A2/NaPi2b	Auristatin F-HPA (antimitotic)	NCT03319628	1
90-yttrium-conjugated FF-21101	RIC	NSCLC, other solid tumors	P-cadherin	Yttrium-90	NCT02454010	1
188-rhenium-conjugated P2045	RIC	SCLC, NSCLC	SSTR2	Rhenium-188	NCT00100256	1, 2
64-cuprum-DOTA-trastuzumab	RIC	Solid tumors	HER2	Cuprum-64	NCT02226276	1
CRLX101	PDC	SCLC, NSCLC, other epithelial cancers	tumor cells	Camptothecin	NCT03531827	1, 2
BT1718	PDC	NSCLC, other solid tumors	MMP14	Maytansinoid mertansine DM1	NCT03486730	1
NKTR102 (Pegylated irinotecan)	PDC	SCLC	tumor cells	Irinotecan	NCT01876446	1
SDX-7320	PDC	NSCLC, other solid tumors	tumor cells	Methionine aminopeptidase 2 (MetAP2) inhibitor	NCT02743637	1
PEN-866	SMDC	SCLC, NSCLC, other epithelial cancers	HSP90	SN-38 (topoisomerase I inhibitor)	NCT03221400	1, 2

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
