# Peer review of "Tumor-Targeted Drug Conjugates as an Emerging Novel Therapeutic Approach in Small Cell Lung Cancer (SCLC)"

_cancers, 2019, doi:10.3390/cancers11091297_

Round 1

Reviewer 1 Report

In this manuscript, Alexander .etc described the tumor specific drug conjugates. In general, the paper is well-written, it focused on small cell lung cancer (SCLC) therapy. Please see below my questions to be addressed before publication:

(1) The authors made a claim of low expression of PD-L1 in SCLC. But the others (T cell therapy is a good example, but a lot more work has been published recently on tuning innate immune system) were missed for example:  CD24, CD47.etc. Can they be used for SCLC, what are the expression levels?

(2) Do ADCs suing DNA-damaging payload still have a future for lung cancer after Rova-T? The choice of payload is crucial? More discussions should be added in this case since we want to learn from the failures.

(3) I think the paper will be more interesting if the authors have a separated section to discuss the ongoing studies about ADCs combined with immunotherapy (any indication on SCLC?).

Author Response

We thank Reviewer #1 for positive feedback, and we respond to specific comments and questions as follows:

Q1: The authors made a claim of low expression of PD-L1 in SCLC. But the others (T cell therapy is a good example, but a lot more work has been published recently on tuning innate immune system) were missed for example:  CD24, CD47.etc. Can they be used for SCLC, what are the expression levels?

Response: We agree with the reviewer that CD24 and CD47 are described in the literature as highly expressed surface molecules on SCLC tumor cells (PMIDs:27373157, 27294525). This point is now mentioned in the revised text, and the existence of potentially promising therapies with ADCs targeting CD47 in other cancers is briefly mentioned, along with some current references (PMID: 27294525). However, antibodies targeting these molecules are not approved for the treatment of SCLC, and there are as yet no preclinical studies or clinical trials accessing efficacy of ADCs with CD24 or CD47-targeting moieties in lung cancer setting. Unfortunately, due in part to strict review length limits and in part due to immune checkpoint inhibitors not being the primary focus of the current review, we are unable to more fully discuss alternative immunotherapy approaches in SCLC.  

Q2: Do ADCs suing DNA-damaging payload still have a future for lung cancer after Rova-T? The choice of payload is crucial? More discussions should be added in this case since we want to learn from the failures.

Response: We believe the primary factors governing efficacy for ADCs are the expression of the antigen (level of expression on tumor, selectivity of expression on tumor versus non-tumor tissue), and properties of the antibody moiety and linker (discussed in detail in paragraphs 2-4 of the section 2.1). We do not think that nature of the payload plays crucial role in ADC efficacy, as long as the payload is efficient in inducing cytotoxicity at low intracellular concentrations.

In addition, we now note in the revised text, Rova-T has not yet been abandoned; it is currently undergoing clinical assessment in SCLC setting along with another ADC, possessing DNA-damaging payload – sacituzumab govitecan. Please see revised paragraphs 2 and 4 of section 2.2.

Q3: I think the paper will be more interesting if the authors have a separated section to discuss the ongoing studies about ADCs combined with immunotherapy (any indication on SCLC?).

Response: Unfortunately, we are not able to address this promising approach in a greater detail than is mentioned in the manuscript, due to the lack of any published data in SCLC setting and review length restrictions.

Reviewer 2 Report

This is a detailed manuscript which describes current tumour-targeting drug conjugates for the treatment of SCLC.  This paper also describes potential tumour-targeting drug conjugates which could be explored for the treatment of SCLC.  While an interesting topic, I have some queries below:

While the use of SMDCs and PDCs have potential to be used to treat SCLC, I think some further clarification is needed. For example, on page 11, the authors mention PS as an SMDC target, but justification for this (e.g. extracellular expression of PS in SCLC) is needed.  And why would PDCs be effective in SCLC?  Is their pre-clinical data in SCLC to support this and also the SMDCs mentioned?  What were the results of the clinical trial of T-DM1 in NSCLC? Different polymers are mentioned in Figure 2, but not all are mentioned in the text, are these relevant?

Minor issues

Line 43, …in a Japanese randomized phase 3 clinical trial….

Line 48, …(ICIs) in the last decade

Sentence 63 needs to be reworded

Line 69, protein-targeted targeted therapies

There’s a total of 5 FDA approved ADCs, 3 targeting solid tumours (Polatuzumab vedotin-piiq recently approved for lymphoma)

Line 104, ‘reduce this concept to practice’ is the wrong wording 

Figure 1, maytansinoids and pyrrolobenzodiazepines spelt wrong

Line 144, …research seeks to optimize..

Line 151, …they and release..

Line 221, …(ORR) of 38%....

Line 293, treatment spelt incorrectly

Figure 2, phosphatidylserines, vinca alkaloids and maytansinoids are spelt incorrectly

Line 337, …highly active, and often selected for, in tumour cells.

Line 359,  Title should be changed to something along the lines of  ‘Clinical perspective of SMDC in SCLC’

Line 402, …two drugas….

Line 421, ..PS was found to be expressed….

Author Response

We thank Reviewer #2 for the overall positive feedback and for all of the useful specific comments. In response:

Q1: While the use of SMDCs and PDCs have potential to be used to treat SCLC, I think some further clarification is needed. For example, on page 11, the authors mention PS as an SMDC target, but justification for this (e.g. extracellular expression of PS in SCLC) is needed.  And why would PDCs be effective in SCLC?  Is their pre-clinical data in SCLC to support this and also the SMDCs mentioned?

Response: We have included appropriate justification, supported by references to the literature (PMID: 25702090), for PS as a target in SCLC.

In our revised manuscript, we also note that although there are not yet extensive data regarding PDCs testing in the SCLC setting, numerous studies (PMIDs: 24647104; 18423779; 24131140; 26615328) report PDCs efficacy in other solid cancers, suggesting that this approach has potential implication in SCLC setting. Moreover, a cyclodextrin-camptothecin PDC is currently undergoing clinical assessment in SCLC (NCT02769962), and there are reported studies demonstrating successful nanoparticle-delivered gene therapy for SCLC, proving that this approach is working. Please, see extended discussion added to the manuscript in section 4. 

Q2: What were the results of the clinical trial of T-DM1 in NSCLC?

Response: This information has been added to the manuscript, in the first paragraph of section 2.2.

Q3: Different polymers are mentioned in Figure 2, but not all are mentioned in the text, are these relevant?

Response: In the updated Figure 2, the polymers assessed in the specific setting of SCLC are highlighted in red, and are specifically mentioned in the text.  Additional polymers of interest for cancer therapy, not yet assessed for SCLC, are not highlighted.

Minor issues...(primarily list of typos or phrasing).

Response: We greatly appreciate the catching multiple small errors, and we addressed all of them in the revised text. Following the reviewer’s comment, we also added information regarding a fifth FDA-approved ADC, polatuzumab vedotin-piiq ADC, that was approved in June 2019.